# GROUNDED-VIDEOLLM: SHARPENING FINE-GRAINED TEMPORAL GROUNDING IN VIDEO LARGE LANGUAGE MODELS

## ABSTRACT

Video Large Language Models (Video-LLMs) have demonstrated remarkable capabilities in coarse-grained video understanding, however, they struggle with fine-grained temporal grounding. In this paper, we introduce *Grounded-VideoLLM*, a novel Video-LLM adept at perceiving and reasoning over specific video moments in a fine-grained manner. We identify that current Video-LLMs have limitations for fine-grained video understanding since they lack effective temporal modeling and timestamp representation. In light of this, we sharpen our model by incorporating (1) an additional temporal stream to encode the relationships between frames and (2) discrete temporal tokens enriched with specific time knowledge to represent timestamps. To optimize the training of *Grounded-VideoLLM*, we employ a multi-stage training scheme, beginning with simple video-captioning tasks and progressively introducing video temporal grounding tasks of increasing complexity. To further enhance *Grounded-VideoLLM*'s temporal reasoning capability, we also curate a grounded VideoQA dataset by an automatic annotation pipeline. Extensive experiments demonstrate that *Grounded-VideoLLM* not only excels in fine-grained grounding tasks such as temporal sentence grounding, dense video captioning, and grounded VideoQA, but also shows great potential as a versatile video assistant for general video understanding.

## 1 INTRODUCTION

Multi-modal Large Language Models (MLLMs) have made remarkable progress in image-level understanding (Liu et al., 2023; Dai et al., 2023; Li et al., 2023a). However, extending their capabilities to the video domain poses distinct challenges. Unlike static images, the temporal nature of videos challenges models to process not only visual content but also the sequence and timing of events. While current Video-LLMs (Xu et al., 2024a; Li et al., 2024; Zhang et al., 2023b; Lin et al., 2023) are capable of capturing global visual semantics and generating coarse-grained captions for short clips, they struggle with fine-grained video understanding (Liu et al., 2024b; Wang et al., 2024d), which requires decomposing the video along the temporal axis to accurately perceive and reason over specific moments, such as subtle actions, transitions, and events that unfold over time.

Previous research efforts (Ren et al., 2024; Huang et al., 2024a; Qian et al., 2024a; Huang et al., 2024b; Guo et al., 2024) have explored *temporal grounding* to improve fine-grained video understanding. However, two main challenges impede their potential for achieving effective *fine-grained temporal grounding*: **(1)** Models like Video-ChatGPT (Maaz et al., 2024b), P-LLaVA (Xu et al., 2024a), and Video-LLAMA (Zhang et al., 2023b) typically sample multiple frames from a video and encode each frame independently using an image encoder, followed by a feature projector (e.g., sliding Q-former (Ren et al., 2024), slot-based token compression (Guo et al., 2024), or visual adapter (Huang et al., 2024a)). This approach focuses primarily on spatial details while potentially neglecting the temporal relationships between frames (Maaz et al., 2024a). **(2)** Current models also struggle with timestamp representation, which is crucial for pinpointing specific moments in time for fine-grained understanding. Models such as TimeChat (Ren et al., 2024) and VTimeLLM (Huang et al., 2024a) represent timestamps as plain texts, for example, `["from 102.3 to 120.1 seconds"]`. Despite being straightforward, this strategy needs to tokenize

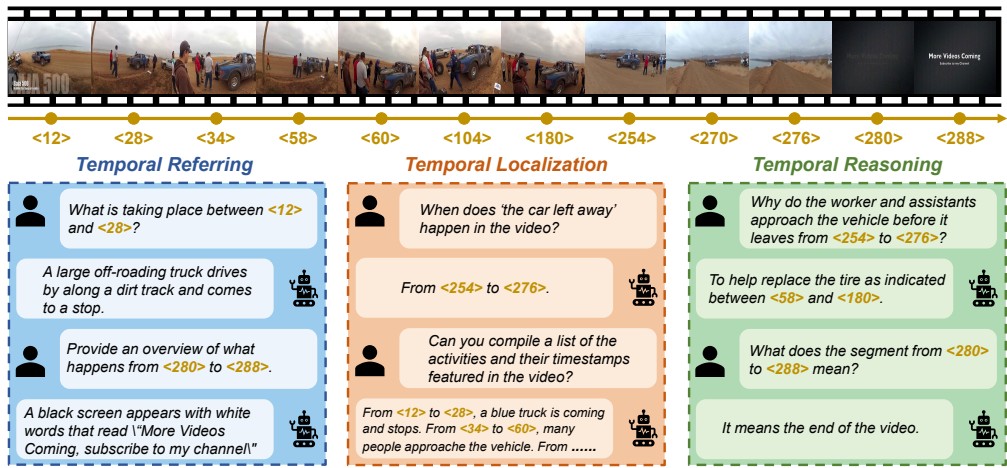

Figure 1: *Grounded-VideoLLM* enables *Temporal Referring/Localizing/Reasoning* for MLLM.

continuous floating-point values, which is inefficient for LLMs since their next-token prediction paradigm struggles with handling numerical data (Schwartz et al., 2024; Frieder et al., 2023).

To further improve video comprehension, we propose to sharpen the model with *fine-grained temporal grounding*, which allows the model to not only recognize *what* happens but also pinpoint *when* it happens in a fine-grained manner. We identify three essential capabilities of *fine-grained temporal grounding*, as illustrated in Figure 1, including *Temporal Referring* (Huang et al., 2024a), *Localizing* (Gao et al., 2017; Caba Heilbron et al., 2015), and *Reasoning* (Xiao et al., 2024; Qian et al., 2024a). *Referring* involves the model precisely describing events within user-specified time intervals. *Localizing* challenges the model to identify the timestamps of a given query and construct a coherent storyline by locating a sequence of events within the video. *Reasoning* combines both *Referring* and *Localizing* capabilities and further requires the ability to answer questions by understanding temporal dynamics and applying wrold knowledge. Achieving these tasks necessitates both effective temporal modeling and robust timestamp representation.

Targeting these goals, we introduce *Grounded-VideoLLM*, a novel Video-LLM that can perceive and reason over specific video moments with fine-grained precision. From the perspective of model architecture, *Grounded-VideoLLM* is built upon two key innovations: (1) **Two-Stream Encoding**: We decompose each segment of the video into spatial and temporal components, and encode each with an expert encoder respectively. The temporal stream extracts motion representations from dense frames and complements the spatial stream which captures appearance representations. This dual-stream approach forms comprehensive video representations enriched with both temporal and spatial information. **(2) Temporal Tokens:** We extend the LLM's vocabulary by introducing discrete tokens specifically crafted for timestamp representation. These temporal tokens denote relative time positions within a video and share a unified embedding space with the LLM. This integration allows *Grounded-VideoLLM* to avoid the inefficiency of tokenizing numerical text, enabling seamless prediction of both timestamps and textual outputs in a single sequence of discrete tokens. From the perspective of training and datasets, given that such a unified model requires extensive pre-training, we start with an image-based MLLM (Microsoft, 2024) as the foundation and adopt a three-stage training strategy to ensure efficiency. We meticulously select different tasks for each stage, and progressively refine the model in a "coarse-to-fine" manner, transitioning from image understanding to video comprehension, and ultimately to fine-grained temporal grounding. Furthermore, we enhance the model's temporal reasoning capability by curating 17K grounded VideoQA (Xiao et al., 2024) samples with the assistance of GPT-4. Extensive experiments demonstrate that *Grounded-VideoLLM* shows promising results over existing Video-LLMs not only in traditional video temporal grounding tasks but also in general video understanding benchmarks.

In summary, we make the following contributions: (1) We present key insights into fine-grained temporal grounding for Video-LLMs and introduce *Grounded-VideoLLM*, a Video-LLM adept at perceiving and reasoning over specific video moments. This is achieved through a two-stream architecture for effective temporal modeling and the temporal tokens for efficient timestamp repre-

sentation. (2) We propose a step-by-step training strategy that progressively adapts an image-based MLLM into a robust Video-LLM, while curating a grounded VideoQA dataset to further enhance the temporal reasoning capability. (3) We conduct extensive experiments across various tasks, including temporal sentence grounding, dense video caption, VideoQA, and general Video-LLM benchmarks, demonstrating the superiority of *Grounded-VideoLLM* for fine-grained video understanding.

## 2    RELATED WORK

**Video Large Language Models** have caught a growing interest with the advancements in image-based MLLMs (Zhang et al., 2023b; Lin et al., 2023; Maaz et al., 2024a; Luo et al., 2023). Despite promising results, current Video-LLMs, such as P-LLaVA (Xu et al., 2024a) and Video-ChatGPT (Maaz et al., 2024b), often struggle with temporal understanding (Liu et al., 2024b) and exhibit temporal hallucination (Wang et al., 2024d) when answering questions about specific video moments. These models encode each frame independently using a pre-trained image encoder, concatenating the frame embeddings to create a video representation. This late fusion method can result in video representations that lack inherent temporal information and heavily rely on the position embeddings of LLM (Su et al., 2024) for temporal understanding, limiting the model's capability to perform fine-grained temporal grounding and understanding. In contrast to these studies, we employ a two-stream architecture that integrates a video expert to extract motion features to complement the appearance features during the early encoding process. Additionally, we employ a progressive training strategy that gradually adapts an image-based MLLM for fine-grained video understanding. Unlike concurrent studies such as SlowFast-LLaVA (Xu et al., 2024b) and VideoGPT+ (Maaz et al., 2024a), which also utilize a two-stream architecture, we specifically targets fine-grained temporal grounding through a unique encoding/pooling/training strategy for dense frames and grounding design.

**Video Temporal Grounding (VTG)** aims to associate specific video moments with their corresponding timestamps. Traditional VTG tasks include Temporal Sentence Grounding (Gao et al., 2017; Hendricks et al., 2018) and Dense Video Captioning (Caba Heilbron et al., 2015; Zhou et al., 2018). Other tasks, such as Grounded VideoQA (Xiao et al., 2024), emphasize reasoning over videos with grounded information. Given the emerging reasoning capabilities of Video-LLMs, many studies have investigated how to adapt them for VTG tasks. For example, TimeChat (Ren et al., 2024), VTimeLLM (Huang et al., 2024a), and LITA (Huang et al., 2024b) perform temporal grounding using a fully text-to-text approach through instruction-tuning datasets. Momentor (Qian et al., 2024a) introduces a temporal perception module to address the quantization errors associated with time tokens, while VTG-LLM (Guo et al., 2024) incorporates a limited set of absolute-time tokens to handle timestamp knowledge. Compared to these studies, we avoid textual representation of timestamps and instead introduce discrete and relative temporal tokens for more efficient timestamp encoding. While Vid2Seq (Yang et al., 2023) also employs specialized tokens to indicate temporal positions, it relies heavily on large-scale pre-training from scratch using noisy transcribed speech and is limited to dense video captioning. Our two-stream architecture and progressive training strategy, however, enable MLLMs to efficiently comprehend videos and effectively handle diverse tasks of fine-grained temporal referring, localizing, and reasoning.

## 3    MODEL ARCHITECTURE

Given that current MLLMs already exhibit strong image-understanding capabilities, our architecture aims to sharpen temporal awareness by capturing motion dynamics across frames, which serve as a vital supplement to spatial content. As shown in Figure 2, we develop *Grounded-VideoLLM* upon a well-established MLLM for spatial comprehension and integrate an expert video encoder for temporal comprehension. Additionally, to avoid tokenizing numerical texts, we incorporate temporal tokens into the LLM's vocabulary for efficient and unified timestamp representation.

### 3.1    TWO-STREAM ENCODING

To effectively model the long-range temporal structure, given a video $\mathcal{V}$ with $T$ frames, we divide it into $K$ segments and employ a segment-wise encoding strategy (Wang et al., 2016). Due to the inherent redundancy of consecutive frames in videos, each segment can be naturally represented from two perspectives: spatial and temporal. The spatial representation of each segment is derived

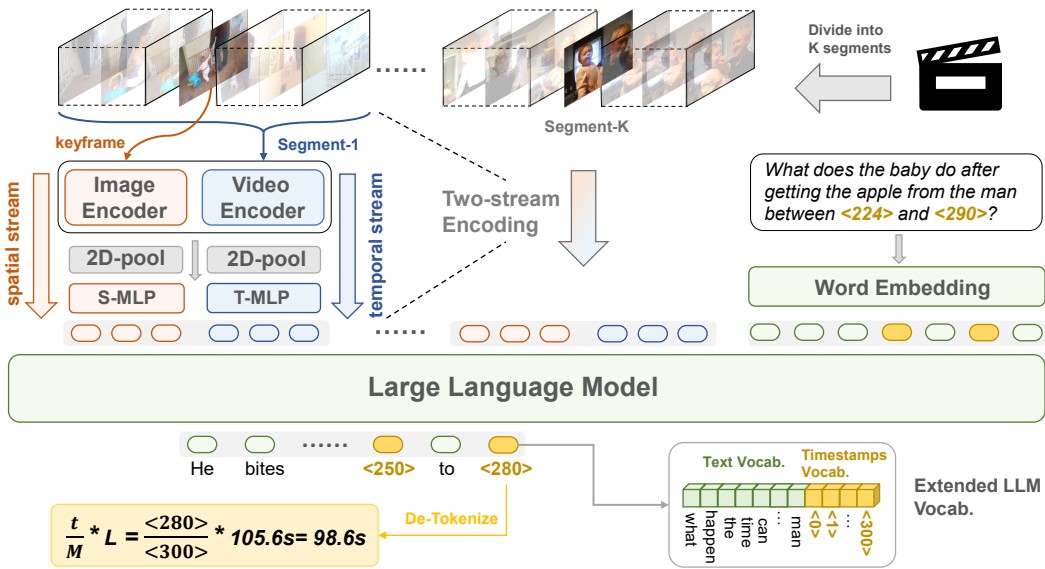

Figure 2: Overview of *Grounded-VideoLLM*. For temporal modeling, we employ a segment-wise encoding strategy by decomposing each segment into a spatial part and a temporal part and encoding each respectively. For timestamp representation, we introduce additional special temporal tokens sharing a unified embedding space with LLM.

from an individual keyframe, capturing the primary appearance semantics, while the temporal representation is learned from multiple frames depicting the motion evolution within the segment. We discuss the details of our two-stream segment encoding as follows.

**Spatial Stream.** We sample the middle frame from each segment as the keyframe and extract its spatial features using the original image encoder from the MLLM (Radford et al., 2021), resulting in spatial features $\mathbf{F}_S \in \mathbb{R}^{H_S \times W_S \times D_S}$, where $H_S, W_S, D_S$ denote the height, width and dimension of the spatial features. Since dense frames are crucial for fine-grained temporal grounding, an appropriate pooling strategy is required to reduce token length for efficient computation. As indicated by Xu et al. (2024a) and Yao et al. (2024) that a 2D average pooling is both efficient and robust for spatial downsampling, we employ a 2D pooling kernel with a size $\sigma_S \times \sigma_S$ over the feature map and gets $\mathbf{F}_S \in \mathbb{R}^{N_S \times D_S}$ as the feature for spatial stream, where $N_S = \frac{H_S}{\sigma_S} \times \frac{W_S}{\sigma_S}$.

**Temporal Stream.** Traditional two-stream networks (Simonyan & Zisserman, 2014; Feichtenhofer et al., 2016) typically encode the optical flow as the temporal stream. However, given the scale of data and parameters involved in MLLMs, extracting optical flow is computationally expensive and impractical. Consequently, we resort to a strong and well pre-trained video encoder, InternVideo2 (Wang et al., 2024b), to extract motion representations for each segment, using a lower resolution but more frames. Specifically, we input each segment, containing $\frac{T}{K}$ frames, into the video encoder to obtain the segment-level features $\mathbf{F}_T \in \mathbb{R}^{\frac{T}{K} \times H_T \times W_T \times D_T}$, where $H_T, W_T, D_T$ denote the height, width and dimension of each frame feature. Similar to the spatial stream, we apply a 2D average pooling strategy to downsample $\mathbf{F}_T$. However, as the temporal stream focuses primarily on temporal modeling, we retain the complete temporal information by only pooling along the spatial dimensions. Specifically, we aggressively downsample $\mathbf{F}_T$ using a kernel with a larger size of $\sigma_T \times \sigma_T$, resulting in the compressed $\mathbf{F}_T \in \mathbb{R}^{\frac{T}{K} \times N_T \times D_T}$ for temporal stream, where $N_T = \frac{H_T}{\sigma_T} \times \frac{W_T}{\sigma_T}$.

To get the complete segment-level representation, we flatten the features of the spatial stream and temporal stream and concat them together :

$$\mathbf{F}_{Seg} = \text{Concat}\left[\text{Flatten}(f(\mathbf{F}_S)); \text{Flatten}(g(\mathbf{F}_T))\right], \mathbf{F}_{Seg} \in \mathbb{R}^{(N_S + \frac{T}{K} \cdot N_T) \times D} \quad (1)$$

where $f(\cdot)$ and $g(\cdot)$ are two MLPs that project the visual features to LLM's dimension $D$. The final video representation is formed by concatenating the $K$ segment-level representations $\mathbf{F}_{Seg}$, resulting in $\mathbf{F}_{Vid} \in \mathbb{R}^{K \cdot (N_S + \frac{T}{K} \cdot N_T) \times D}$. This representation retains detailed spatial information across all

segments along with their global temporal contexts, while maintaining a manageable token length. The combined video representation $\mathbf{F}_{Vid}$ is then fed into the LLM serving as soft prompts, alongside the word embeddings of the instruction text $\mathbf{F}_{Text}$ to generate the target response $\mathcal{A}$. The model is trained using the cross-entropy loss function with trainable parameters $\theta$:

$$\mathcal{L} = -\sum_{t=1}^{L_a} log P_\theta(\mathcal{A}_t | \mathcal{A}_{<t}, \mathbf{F}_{Vid}, \mathbf{F}_{Text}) \tag{2}$$

where $\mathcal{A}_t$ is predicted autoregressively at position $t$, and $L_a$ is the sequence length of the ground truth answer text $\mathcal{A}$.

### 3.2 Unified Temporal Tokens

Given a text span depicting a particular video clip and its associated timestamps, we employ a relative time representation that converts continuous timestamps into a sequence of discrete temporal tokens. For a video $\mathcal{V}$ with a duration of $L$ seconds, we evenly divide $\mathcal{V}$ into $M$ equal-length chunks, and then define $M + 1$ anchor points (ranging from `<0>` to `<M>`) across $\mathcal{V}$, representing relative temporal positions. Each anchor point corresponds to a specific timestamp of $\mathcal{V}$ and is encoded as a temporal token. For instance, `<0>` denotes the very start of $\mathcal{V}$ while `<M>` indicates the end. These $M + 1$ tokens are added to the LLM's vocabulary to enable unified modeling alongside text. A specific continuous timestamp $\tau$ can be easily converted to a temporal token `<t>` and vice verse:

$$t = \text{Round}(M \cdot \frac{\tau}{L}), \quad \tau = L \cdot \frac{t}{M} \tag{3}$$

While this may introduce minor quantization errors, these can be minimized by selecting an appropriate $M$. We then organize the text span and its corresponding temporal tokens in a unified format. Both text tokens and temporal tokens are mapped to embeddings through the extended word embedding layer of LLM. For example, one input representation is as follows:

> `<video>`$\mathbf{F}_{Vid}$`</video> <grounded>` From `<0>` to `<6>`, a baby is crying. From `<7>` to `<16>`, a man is coming and picking up the baby. From `<20>` to `<25>`, the baby is eating an apple. From `<27>` to `<35>`, the baby is smiling happily.``

where `` and `` indicate start- and end-of-sequence, `<video>` and `</video>` represent the beginning and end of encoded video representations. `<grounded>` is a special token to tell the model should output the grounded timestamps. This strategy avoids the need to tokenize and process numerical values, which has been identified as a limitation of LLMs (Schwartz et al., 2024), and it greatly simplifies the representation of timestamps within the unified embedding space of LLMs. Consequently, text and timestamps can be jointly decoded as a single sequence, following the objective outlined in Eq. (2).

## 4 Progressive Training

Different from previous methods (Zhang et al., 2023b; Lin et al., 2023) that train models from scratch using mixed image and video datasets, we start with a pre-trained image-based MLLM (Microsoft, 2024) and progressively enhance its fine-grained temporal grounding capabilities. This strategy can be applied to any off-the-shelf MLLM and is more efficient. Table 1 enumerates the datasets used at different training stages, and Table 9 lists the prompts for different tasks.

**Stage-1: Video-Caption Alignment.** Feature alignment is widely used to improve training efficiency (Liu et al., 2023). In this stage, we leverage approximately 1.28 million video-text pairs sampled from diverse sources (Wang et al., 2024a; Bain et al., 2021; Chen et al., 2024b) to align the video encoder's features with the MLLM. This alignment allows the MLLM, which was pre-trained solely on images, to gain a foundational understanding of videos. Since 2D down-sampling has been applied to the visual features, only the two projectors $f(\cdot)$ and $g(\cdot)$ are set to be trainable, while the video encoder, image encoder, and LLM remain frozen. As this stage does not involve any video temporal grounding tasks, the temporal tokens described in Sec.3.2 are not yet incorporated.

Table 1: Datasets used at three training stages. Tasks with gray background consist of datasets regarding temporal grounding.

| Training Stage | Task | # of Samples | Datasets |
|---|---|---|---|
| Video-Caption Alignment | Video Captioning | 1.28M | WebVid-10M, Panda-70M, InternVid-10M |
| Temporal Token Alignment | Temporal Sentence Grounding | 149K | VTimeLLM-Stage2 |
| | Dense Video Captioning | 92K | VTimeLLM-Stage2, Moment-10M, InternVid-G |
| | Temporal Referring | 95K | VTimeLLM-Stage2, InternVid-G |
| Multi-Task Instruction Tuning | Grounded Conversation | 442K | ANet-RTL, Moment-10M |
| | Temporal Sentence Grounding | 84K | DiDeMo, HiREST, QuerYD, VTG-IT |
| | Dense Video Caption | 41K | COIN, ViTT, YouCook2, VTG-IT |
| | Grounded VideoQA | 17K | Self Collected |
| | Converstation | 233K | VCG-Plus-112K, Videochatgpt-100K, Videochat2-Conv |
| | VideoQA | 282K | EgoQA, NExT-QA, Intent-QA, STAR, CLEVRER, WebVid-QA |
| | Classification | 66K | SthSthV2, Kinetics |
| | Video Captioning | 136K | TextVR, YouCook2, WebVid, ShareGPT4Video |

**Stage-2: Temporal Token Alignment.** While video-caption alignment effectively connects videos and the MLLM at a coarse semantic level, a gap persists between this alignment and fine-grained temporal grounding. To address this, we introduce the temporal tokens described in Sec.3.2 and continue pre-training the model on a diverse range of grounding datasets (Huang et al., 2024a; Qian et al., 2024a; Wang et al., 2024c), focusing on tasks such Temporal Sentence Grounding, Dense Video Captioning, and Temporal Referring, which enables the model to refer to and localize temporal information effectively. Since new tokens are introduced, the trainable parameters in this stage include the two projectors, $f(\cdot)$ and $g(\cdot)$, the word embedding matrix, and the final classifier head of the LLM. This step enhances the model's ability to comprehend multiple events and aligns the temporal tokens with both the video timelines and the LLM's semantic space.

**Stage-3: Multi-Task Instruction Tuning.** Following the initial two stages of pre-training, the model has developed a basic understanding of video content and the ability to refer to and locate specific timestamps. In this stage, we will further enhance the model's fine-grained temporal grounding while improving its responsiveness to diverse user instructions. To achieve this, we gather two types of datasets: (1) We compile a wide range of public datasets for video temporal grounding tasks, similar to Time-IT (Ren et al., 2024) and VTG-IT (Guo et al., 2024), which include tasks of dense video captioning, temporal sentence grounding, and grounded VideoQA. (2) We incorporate a selection of instructional video-to-text datasets from VideoChat2 (Li et al., 2024), which feature tasks such as conversations, classification, question answering, and captioning. Additionally, we include ShareGPT-4Video (Chen et al., 2024a) to further enhance the model's ability to generate detailed video captions. By utilizing these diverse datasets, which encompass both temporal grounding and video instruction tasks, *Grounded-VideoLLM* excels in temporal referring, localization, reasoning, and general comprehension of video content. In this stage, the trainable parameters remain the same as in Stage 2, with the addition of LoRA parameters (Hu et al., 2022) for the LLM.

## 5 GROUNDED VIDEOQA DATASET GENERATION

Grounded VideoQA requires the model to not only answer questions but also identify relevant video moments that support predicted answers, thereby demonstrating the model's temporal reasoning abilities. The NExT-GQA dataset (Xiao et al., 2024) was manually developed by extending NExT-QA (Xiao et al., 2021) with temporal labels for start and end timestamps. However, annotating these temporal labels is labor-intensive and time-consuming, which has limited NExT-GQA only to QA pairs for the validation and test sets. To create a scalable training dataset, we utilized OpenAI GPT-4 (Achiam et al., 2023) to assist in constructing training sets for the grounded VideoQA task. These sets were built on public datasets that already contain temporal labels, such as ActivityNet-Caption (Caba Heilbron et al., 2015) and QVHighlights (Lei et al., 2021). We framed the task as a multiple-choice problem using a two-round conversational format, as depicted in Figure 3.

Specifically, we input event descriptions along with their timestamps into GPT-4 and prompted it to generate corresponding question-answer pairs, as shown in Table 8. To create distractor options for the multiple-choice questions, we retrieved the top 50 questions most similar to the generated question, based on cosine similarity using an embedding model (Reimers, 2019). The answers to these 50 retrieved questions served as candidates for distractors. From this pool, we randomly sampled

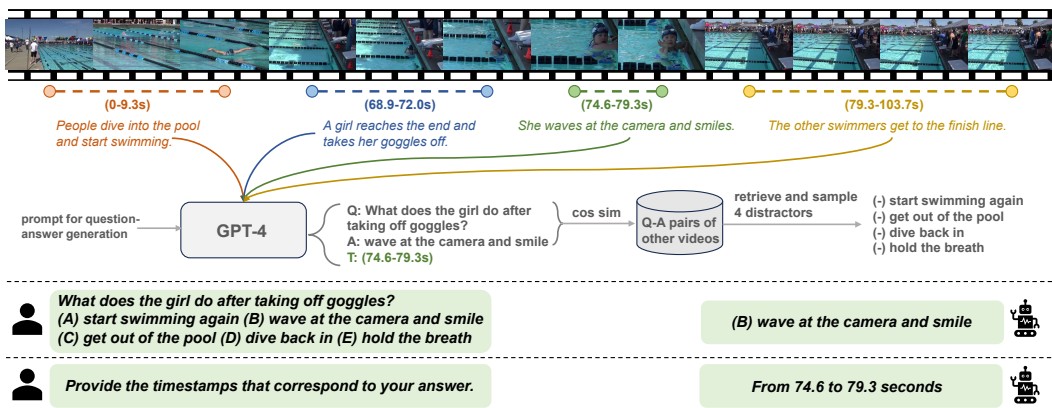

Figure 3: Examples of annotation pipeline and generated data for Grounded VideoQA.

four distractors with cosine similarities to the correct answer ranging from 0.2 to 0.9, ensuring that the distractors were contextually relevant but not overly similar to the correct answer. The ground-truth timestamps for answering each question were derived from the timestamps of the associated event descriptions. The constructed dataset comprises 17K samples, which have been incorporated into the training sets for Stage 3, further enhancing the model's temporal reasoning performance.

# 6 EXPERIMENTS

## 6.1 EXPERIMENT SETTING

**Implementation Details.** We select the Phi3.5-Vision-Instruct-3.8B (Microsoft, 2024) as the base MLLM for our *Grounded-VideoLLM*. For temporal stream encoding, we adopt InternVideo2-1B (Wang et al., 2024b) as the video encoder. Each video is sampled as a sequence of $T = 96$ frames, which are evenly divided into $K = 12$ segments. We set the pooling size $\sigma_S = 2$ for the spatial stream while $\sigma_T = 4$ for the temporal stream respectively, which results in $N_S = 144$ tokens per frame for the spatial stream while $N_T = 16$ tokens per frame for the temporal stream. Moreover, we introduce $M = 300$ temporal tokens into the LLM's vocabulary for timestamp representation. More implementation details can be found in Appendix A.1.

**Tasks and Benchmarks.** To thoroughly evaluate *Grounded-VideoLLM* in fine-grained temporal grounding, we assess it across three video temporal grounding tasks: *Temporal Sentence Grounding*, *Dense Video Captioning*, and *Grounded VideoQA*, utilizing datasets such as **Charades-STA** (Gao et al., 2017), **ActivityNet-Captions** (Caba Heilbron et al., 2015), and **NExT-GQA** (Xiao et al., 2024). We also show its reasoning capability by the task of Open-Ended VideoQA with datasets including **MSVD-QA**, **MSRVTT-QA** (Xu et al., 2017), and **ActicityNet-QA** (Yu et al., 2019). Additionally, to evaluate the model's general video understanding capabilities, we benchmark *Grounded-VideoLLM* against existing models using modern Video-LLM benchmarks including **VCG-Bench** (Maaz et al., 2024b) and **MVBench** (Li et al., 2024).

**Evaluation Metrics.** For temporal sentence grounding, we report the metric of Intersection over Union (**IoU**) (Gao et al., 2017) between the timestamps predicted by the model and the ground truth, including **Recall at IoU** thresholds of {0.3, 0.5, 0.7} and their **mean IoU**. For dense video captioning, we use metrics including **SODA_c** (Fujita et al., 2020) which is specifically tailored for the video's storyline, and **METEOR** score (Banerjee & Lavie, 2005), which is the average of traditional METEOR scores that are calculated based on matched pairs between generated events and the ground truth across IoU thresholds of {0.3, 0.5, 0.7, 0.9}. For Visually-grounded VideoQA, we calculate both the Intersection of Prediction (**IoP**) (Xiao et al., 2024) and Intersection of Union (**IoU**), and use **Acc@GQA** (Xiao et al., 2024) to measure the percentage of questions that are both correctly answered and visually grounded with IoP ≥ 0.5. The responses of Open-Ended VideoQA and VCG-Bench are evaluated by GPT-3.5 with the prompts introduced by Video-ChatGPT (Maaz et al., 2024b). More evaluation details are in Appendix A.3.

Table 2: Zero-shot results on temporal sentence grounding and dense video captioning tasks.

| Model | LLM Scale | Charades-STA | | | | ActivityNet-Grounding | | | | ActivityNet-Captions | |
|---|---|---|---|---|---|---|---|---|---|---|---|
| | | R@0.3 | R@0.5 | R@0.7 | mIoU | R@0.3 | R@0.5 | R@0.7 | mIoU | SODA_c | METEOR |
| Video-LLaMA (Zhang et al., 2023b) | 7B | 25.2 | 10.6 | 3.4 | 16.8 | 21.9 | 10.8 | 4.9 | 16.5 | 1.9 | 1.9 |
| SeViLA (Yu et al., 2023) | 3B | 27.0 | 15.0 | 5.8 | 18.3 | 31.6 | 19.0 | 10.1 | 23.0 | - | - |
| Video-ChatGPT (Maaz et al., 2024b) | 7B | 27.2 | 6.2 | 1.9 | 19.7 | 19.5 | 10.6 | 4.8 | 14.2 | 1.9 | 2.1 |
| Valley (Luo et al., 2023) | 7B | 28.4 | 1.8 | 0.3 | 21.4 | 30.6 | 13.7 | 8.1 | 21.9 | 0.3 | 0.8 |
| VideoChat2 (Li et al., 2024) | 7B | 38.0 | 14.3 | 3.8 | 24.6 | 40.8 | 27.8 | 9.3 | 27.9 | - | - |
| VideoChat (Li et al., 2023b) | 7B | 32.8 | 8.6 | 0.0 | 25.9 | 23.5 | 12.6 | 6.0 | 17.4 | 0.9 | 0.9 |
| Momenter (Qian et al., 2024a) | 7B | 42.6 | 26.6 | 11.6 | 28.5 | 42.9 | 23.0 | 12.4 | 29.3 | 2.3 | 4.7 |
| VTimeLLM (Huang et al., 2024a) | 7B | 51.0 | 27.5 | 11.4 | 31.2 | 44.0 | 27.8 | 14.3 | 30.4 | 5.8 | 6.8 |
| TimeChat (Ren et al., 2024) | 7B | - | 32.2 | 13.4 | - | - | - | - | - | - | - |
| VTG-LLM (Guo et al., 2024) | 7B | - | 33.8 | 15.7 | - | - | - | - | - | - | - |
| HawkEye (Wang et al., 2024c) | 7B | 50.6 | 31.4 | 14.5 | 33.7 | 49.1 | 29.3 | 10.7 | 32.7 | - | - |
| Grounded-VideoLLM | 4B | 54.2 | 36.4 | 19.7 | 36.8 | 46.2 | 30.3 | 19.0 | 36.1 | 6.0 | 6.8 |

Table 3: Results on NExT-GQA. Acc@GQA is defined as the percentage of questions that are both correctly answered and visually grounded with IoP $\geq$ 0.5.

| Model | Acc@GQA | mIoP | IoP@0.3 | IoP@0.5 | mIoU | IoU@0.3 | IoU@0.5 |
|---|---|---|---|---|---|---|---|
| VIOLETv2 (Fu et al., 2023) | 12.8 | 23.6 | 25.1 | 23.3 | 3.1 | 4.3 | 1.3 |
| Temp[CLIP] NG+ (Xiao et al., 2024) | 16.0 | 25.7 | 31.4 | 25.5 | 12.1 | 17.5 | 8.9 |
| SeViLA (Yu et al., 2023) | 16.6 | 29.5 | 34.7 | 22.9 | 21.7 | 29.2 | 13.8 |
| HawkEye (Wang et al., 2024c) | - | - | - | - | 25.7 | 37.0 | 19.5 |
| LangRepo (Kahatapitiya et al., 2024) | 17.1 | 31.3 | - | 28.7 | 18.5 | - | 12.2 |
| FrozenBiLM NG+ (Yang et al., 2022) | 17.5 | 24.2 | 28.5 | 23.7 | 9.6 | 13.5 | 6.1 |
| VideoStreaming (Qian et al., 2024b) | 17.8 | 32.2 | - | 31.0 | 19.3 | - | 13.3 |
| LLoVi (Zhang et al., 2023a) | 24.3 | 37.3 | - | 36.9 | 20.0 | - | 15.3 |
| Grounded-VideoLLM | 26.7 | 34.5 | 42.6 | 34.4 | 21.1 | 30.2 | 18.0 |

## 6.2 MAIN RESULTS

**Temporal Sentence Grounding** requires the model to identify the precise time interval corresponding to a given query sentence. As shown in Table 2, *Grounded-VideoLLM* achieves performance on "mIoU" with 36.8 for the Charades-STA (Gao et al., 2017) and 36.1 for ActivityNet-Grounding (Caba Heilbron et al., 2015) respectively, surpassing previous SoTA end-to-end Video-LLMs, i.e., HawkEye (Wang et al., 2024c), by a significant margin (+3.4). It is worth mentioned that the promising performance of "mIoU" are largely attributed to the signigicant gains in terms of "R@0.7" compared with other thresholds, demonstrating that *Grounded-VideoLLM* is more advanced in localizing specific moments within videos with finer granularity.

**Dense Video Captioning** involves generating descriptions for all events in a video, along with their corresponding start and end timestamps. We evaluated *Grounded-VideoLLM* on the ActivityNet-Captions (Caba Heilbron et al., 2015), and the results in Table 2 show that our method achieves the highest SODA_c score (6.0), which demonstrates that, thanks to the Temporal Token Alignment training stage, *Grounded-VideoLLM* is highly effective in identifying the multi-event structure of the video and capturing complete storylines. The highest METEOR score (6.8) also indicates that *Grounded-VideoLLM* provides more detailed event descriptions compared with other Video-LLMs.

**NExT-GQA** (Xiao et al., 2024) is quite challenging since it requires the model to not only correctly answer questions but also provide timestamps that support the answers, highlighting the temporal reasoning capability. According to Table 3, *Grounded-VideoLLM* achieves the highest Acc@GQA (26.7, +2.4) and delivers comparable IoU and IoP scores to models such as SeViLA (Yu et al., 2023) and LLoVi (Zhang et al., 2023a), which use specialized grounding modules or rely on proprietary large language models (Achiam et al., 2023). The highest Acc@GQA score further demonstrates *Grounded-VideoLLM*'s capability in both fine-grained temporal grounding and high-level reasoning.

**Open-Ended VideoQA.** As shown in Table 4, *Grounded-VideoLLM* achieves state-of-the-art or comparative performance across MSVD-QA, MSRVTT-QA (Xu et al., 2017), and ActivityNet-QA (Yu et al., 2019), highlighting its advancements in general video question answering.

**General Video-LLM Benchmarks.** While *Grounded-VideoLLM* excels in fine-grained temporal grounding, we aim to ensure that it maintains robust performance in general video understanding. Therefore, we conducted a comprehensive evaluation using VCG-Bench (Maaz et al., 2024b) and

Table 4: Results on zero-shot Open-Ended VideoQA and VCG-Bench. VCG-Bench contains five aspects: Correctness of Information (CI), Detail Orientation (DO), Contextual Understanding (CU), Temporal Understanding (TU), and Consistency (CO).

| Model | MSVD-QA | | MSRVTT-QA | | ANet-QA | | VCG-Bench | | | | | |
|---|---|---|---|---|---|---|---|---|---|---|---|---|
| | Acc. | Score | Acc. | Score | Acc. | Score | CI | DO | CU | TU | CO | Avg. |
| *Video-LLMs w/o temporal grounding capability.* | | | | | | | | | | | | |
| Video-LLaMA (Zhang et al., 2023b) | 51.6 | 2.5 | 29.6 | 1.8 | 12.4 | 1.1 | 1.96 | 2.18 | 2.16 | 1.82 | 1.79 | 1.98 |
| Video-ChatGPT (Maaz et al., 2024b) | 64.9 | 3.3 | 49.3 | 2.8 | 35.2 | 2.7 | 2.50 | 2.57 | 2.69 | 2.16 | 2.20 | 2.42 |
| Vista-LLaMA (Ma et al., 2024) | 65.3 | 3.6 | 60.5 | 3.3 | 48.3 | 3.3 | 2.44 | 2.64 | 3.18 | 2.26 | 2.31 | 2.57 |
| MovieChat (Song et al., 2024) | 75.2 | 3.8 | 52.7 | 2.6 | 45.7 | 3.4 | 2.76 | 2.93 | 3.01 | 2.24 | 2.42 | 2.67 |
| LongVLM (Weng et al., 2024) | 70.0 | 3.8 | 59.8 | 3.3 | 47.6 | 3.3 | 2.76 | 2.86 | 3.34 | 2.39 | 3.11 | 2.89 |
| VideoChat2 (Li et al., 2024) | 70.0 | 3.9 | 54.1 | 3.3 | 49.1 | 3.3 | 3.02 | 2.88 | 3.51 | 2.66 | 2.81 | 2.98 |
| Chat-UniVi (Jin et al., 2024) | 65.0 | 3.6 | 54.6 | 3.1 | 45.8 | 3.2 | 2.89 | 2.91 | 3.46 | 2.89 | 2.81 | 2.99 |
| P-LLaVA-7B (Xu et al., 2024a) | **76.6** | **4.1** | 62.0 | 3.5 | 56.3 | **3.5** | 3.21 | 2.86 | 3.62 | 2.33 | 2.93 | 3.12 |
| ST-LLM (Liu et al., 2024a) | 74.6 | 3.9 | **63.2** | 3.4 | 50.9 | 3.3 | 3.23 | 3.05 | **3.74** | 2.93 | 2.81 | 3.15 |
| VideoGPT+ (Maaz et al., 2024a) | - | - | - | - | - | - | 3.27 | **3.18** | **3.74** | 2.83 | **3.39** | **3.28** |
| *Video-LLMs w/ temporal grounding capability.* | | | | | | | | | | | | |
| Momentor (Qian et al., 2024a) | 68.9 | 3.6 | 55.6 | 3.0 | 40.8 | 3.2 | - | - | - | - | - | - |
| VTimeLLM (Huang et al., 2024a) | - | - | - | - | - | - | 2.78 | 3.10 | 3.40 | 2.49 | 2.47 | 2.85 |
| LITA (Huang et al., 2024b) | - | - | - | - | - | - | 2.94 | 2.98 | 3.43 | 2.68 | 3.19 | 3.04 |
| *Grounded-VideoLLM* | 76.3 | **4.1** | 60.3 | **3.6** | **56.8** | 3.5 | **3.34** | 2.94 | 3.66 | **3.12** | 3.14 | 3.24 |

Table 5: Results on MVBench multi-choice question answering.

| Model | Avg. | AS | AP | AA | FA | UA | OE | OI | OS | MD | AL | ST | AC | MC | MA | SC | FP | CO | EN | ER | CI |
|---|---|---|---|---|---|---|---|---|---|---|---|---|---|---|---|---|---|---|---|---|---|
| VideoChatGPT (Maaz et al., 2024b) | 32.7 | 23.5 | 26.0 | 62.0 | 22.5 | 26.5 | 54.0 | 28.0 | 40.0 | 23.0 | 20.0 | 31.0 | 30.5 | 25.5 | 39.5 | 48.5 | 29.0 | 33.0 | 29.5 | 26.0 | 35.5 |
| VideoLLaMA (Zhang et al., 2023b) | 34.1 | 27.5 | 25.5 | 51.0 | 29.0 | 39.0 | 48.0 | 40.5 | 38.0 | 22.5 | 22.5 | 43.0 | 34.0 | 22.5 | 32.5 | 45.5 | 32.5 | 40.0 | 30.0 | 21.0 | 37.0 |
| VideoChat (Li et al., 2023b) | 35.5 | 33.5 | 26.5 | 56.0 | 33.5 | 40.5 | 53.0 | 40.5 | 30.0 | 25.5 | 27.0 | 48.5 | 35.0 | 20.5 | 42.5 | 46.0 | 26.5 | 41.0 | 23.5 | 23.5 | 36.0 |
| TimeChat (Ren et al., 2024) | 38.5 | 40.5 | 36.0 | 61.0 | 32.5 | 53.0 | 53.5 | 41.5 | 29.0 | 19.5 | 26.5 | 66.5 | 34.0 | 20.0 | 43.5 | 42.0 | 36.5 | 36.0 | 29.0 | 35.0 | 35.0 |
| Video-LLaVA (Lin et al., 2023) | 43.0 | 46.0 | 42.5 | 56.5 | 39.0 | 53.5 | 53.0 | 48.0 | 41.0 | 29.0 | 31.5 | 82.5 | 45.0 | 26.0 | 53.0 | 41.5 | 33.5 | 41.5 | 27.5 | 38.5 | 31.5 |
| P-LLaVA-7B (Xu et al., 2024a) | 46.6 | 58.0 | 49.0 | 55.5 | 41.0 | 61.0 | 56.0 | 61.0 | 36.0 | 23.5 | 26.0 | 82.0 | 39.5 | 42.0 | 52.0 | 45.0 | 42.0 | 53.5 | 30.5 | 48.0 | 31.0 |
| VideoChat2 (Li et al., 2024) | 51.1 | 66.0 | 47.5 | 83.5 | 49.5 | 60.0 | 58.0 | 71.5 | 42.5 | 23.0 | 23.0 | 88.5 | 39.0 | 42.0 | 58.5 | 44.0 | 49.0 | 36.5 | 35.0 | 40.5 | 65.5 |
| ShareGPT4Video (Chen et al., 2024a) | 51.2 | 49.5 | 39.5 | 79.5 | 40.0 | 54.5 | 82.5 | 54.5 | 32.5 | 50.5 | 41.5 | 84.5 | 35.5 | 62.5 | 75.0 | 51.0 | 25.5 | 46.5 | 28.5 | 39.0 | 51.5 |
| ST-LLM (Liu et al., 2024a) | 54.9 | 66.0 | 53.5 | 84.0 | 44.0 | 58.5 | 80.5 | 73.5 | 38.5 | 42.5 | 31.0 | 86.5 | 36.5 | 56.5 | 78.5 | 43.0 | 44.5 | 46.5 | 34.5 | 41.5 | 58.5 |
| VideoGPT+ (Maaz et al., 2024a) | 58.7 | 69.0 | 60.0 | 83.0 | 48.5 | 66.5 | 85.5 | 75.5 | 36.0 | 44.0 | 34.0 | 89.5 | 39.5 | 71.0 | 90.5 | 45.0 | 53.0 | 50.0 | 29.5 | 44.0 | 60.0 |
| *Grounded-VideoLLM* | 59.4 | 76.0 | 75.5 | 77.0 | 48.0 | 67.5 | 85.5 | 77.0 | 34.5 | 39.5 | 59.5 | 86.5 | 44.5 | 60.5 | 79.0 | 51.5 | 49.0 | 46.0 | 35.0 | 42.5 | 54.0 |

**MVBench (Li et al., 2024).** As shown in Table 4, *Grounded-VideoLLM* achieves promising results in VCG-Bench, with an average score of 3.24, outperforming other Video-LLMs *with temporal grounding capabilities* (e.g., LITA, VTimeLLM). Notably, *Grounded-VideoLLM* surpasses all other Video-LLMs on the TU (Temporal Understanding) task, with a score of 3.12 (+7%), demonstrating its superior temporal understanding, which can be attributed to the two-stream architecture that can capture motion dynamics. For MVBench which provides 4,000 QA pairs spanning a wide range of scenes categorized into 20 fine-grained tasks, the results, presented in Table 5, show that *Grounded-VideoLLM* achieves an average score of 59.4, surpassing other Video-LLMs. Notably, it achieves top performance in several critical tasks requiring perceiving and reasoning over specific video moments, including Action Sequence (AS), Action Prediction (AP), Action Localization (AL), and State Change (SC), with scores of 76.0 (+10%), 75.0 (+26%), 59.5 (+43%), and 51.5 (+14%), respectively, demonstrating significant advancements on fine-grained video understanding.

## 6.3 IN-DEPTH ANALYSIS

**Two-stream Encoding.** We conduct ablations to our two-stream encoding. Specifically, we set two variants by removing the temporal stream while only retaining the spatial stream, where all frame embeddings are concatenated as the video representation: (1) *w/o temporal-stream (dense)* feeds $T = 96$ frames with a pooling size $\sigma_S = 4$ (36 tokens per frame), resulting in a total of $36 \times 96 = 3456$ tokens. (2) *w/o temporal-stream (sparse)* feeds $T = 24$ frames with a pooling size $\sigma_S = 2$ (144 tokens per frame), also resulting in a total of $144 \times 24 = 3456$ tokens. Both variants have a close number of tokens for video

Table 6: Impact of two-stream encoding and alignment stage.

| Model | C-STA | ANet-G | ANet-Cap | |
|---|---|---|---|---|
| | mIoU | mIoU | SODA_c | METEOR |
| *Grounded-VideoLLM* | 36.8 | 36.1 | 6.0 | 6.8 |
| w/o temporal-stream (sparse) | 30.4 (↓ 6.4) | 28.0 (↓ 8.1) | 4.9 (↓ 1.1) | 5.5 (↓ 1.3) |
| w/o temporal-stream (dense) | 34.3 (↓ 2.5) | 29.2 (↓ 6.9) | 5.4 (↓ 0.6) | 6.2 (↓ 0.6) |
| w/o temporal token alignment | 27.5 (↓ 9.3) | 23.1 (↓ 13.0) | 4.7 (↓ 1.3) | 6.4 (↓ 0.4) |

Figure 4: We visualize the attention weights of the LLM when generating the temporal tokens.

representation compared to our two-stream encoding ($12 \times (144 + \frac{96}{12} \times 16) = 3264$ tokens for $\mathbf{F}_{Vid} \in \mathbb{R}^{K \cdot (N_S + \frac{T}{K} \cdot N_T) \times D}$). Table 6 shows that both variants result in a significant performance drop on temporal grounding tasks. Interestingly, the dense frame variant performs slightly better than the sparse frame variant, suggesting that dense temporal information is more critical for grounding tasks than spatial details. Our two-stream architecture strikes a balance by effectively capturing dense motion dynamics while maintaining essential appearance details.

**Temporal Token Alignment.** We further investigate the role of temporal tokens by ablation of the second training stage, Temporal Token Alignment. Quantitative results in Table 6 reveal a sharp performance drop across all tasks, particularly in temporal sentence grounding tasks. Upon analyzing the outputs, we found that the model often produces time intervals spanning nearly the entire video (e.g., from `<0>` to `<300>`), neglecting the association between specific moments and temporal tokens, which leads to misalignment. Qualitatively, we visualize the attention weights of the last layer in the LLM to demonstrate how temporal tokens attend to corresponding video moments. Details of the visualizations are provided in Appendix A.4. As shown in Figure 4 (a), when generating the temporal token, e.g. `<241>` or `<271>`, the attention weights are higher and more focused on their corresponding video moments. Conversely, in Figure 4 (b), when the model is trained without the Temporal Token Alignment stage, the attention weights of temporal tokens become significantly dispersed across irrelevant moments. This illustrates that our multi-stage training strategy is essential for achieving proper alignment between temporal tokens and the video timeline. We also visualize the embedding distribution of temporal tokens in Appendix A.5.

**Grounded VideoQA Dataset.** We validate the role of our constructed grounded-VideoQA dataset by removing it from the training sets of stage-3. Since the model without training on our dataset usually generates free-form texts when asked to output the timestamps supporting the answer, we reformulate it as a temporal sentence grounding task, where we combine the predicted answer and question into a single sentence and ask the model to localize its timestamps. Table 7 suggests that there is a significant performance decrease with regard to Acc@GQA ($\downarrow$ 6.6), mIoP ($\downarrow$ 12.3), and mIoU ($\downarrow$ 8.2), from which we can conclude that our Grounded VideoQA dataset is essential to further enhance the model's temporal reasoning capability.

Table 7: Impact of grounded VideoQA dataset.

| Model | NExT-GQA | | |
|---|---|---|---|
| | Acc@GQA | mIoP | mIoU |
| *Grounded-VideoLLM* | 26.7 | 34.5 | 21.1 |
| w/o grounded VideoQA | 18.1 ($\downarrow$ 8.6) | 22.2 ($\downarrow$ 12.3) | 12.9 ($\downarrow$ 8.2) |

## 7  CONCLUSION

We present *Grounded-VideoLLM*, a Video-LLM capable of fine-grained perception and reasoning over specific video moments. This is achieved through a novel model architecture that incorporates two-stream encoding for effective temporal modeling, along with the temporal tokens for efficient timestamp representation. We employ a multi-stage training scheme, starting with an image-based MLLM and progressively equipping it with *fine-grained temporal grounding* capabilities. Additionally, we curate a grounded-VideoQA dataset to further enhance the model's temporal reasoning ability. Extensive experiments demonstrate that *Grounded-VideoLLM* not only excels in video temporal grounding tasks but also performs strongly on general video understanding benchmarks.

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

# A APPENDIX

## A.1 MORE IMPLEMENTATION DETAILS

Phi3.5-Vison-Instruct (Microsoft, 2024) consists of a CLIP style ViT image encoder (Radford et al., 2021), an MLP projector $f(\cdot)$, and the large language model Phi3.5-mini-3.8B (Abdin et al., 2024). Each video is sampled as a sequence of $T = 96$ frames, which are evenly divided into $K = 12$ segments. For the spatial stream encoded by the ViT in Phi3.5-V, we adopt a higher resolution 336×336, but a lower resolution 224×224 for the temporal stream encoded by InternVideo-2.

We set the pooling size $\sigma_S$ to be 2 while $\sigma_T$ to be 4 respectively. For the spatial stream, each frame takes up $24 \times 24 = 576$ tokens before while $12 \times 12 = 144$ tokens after pooling. For the temporal stream, each frame takes up $16 \times 16 = 256$ tokens before while $4 \times 4 = 16$ tokens after pooling. Therefore, we have an overall of $K \times (144 + \frac{T}{K} \times 16) = 3264$ tokens in total.

During training, we use the AdamW optimizer with a cosine learning rate decay and set the learning rate as 2e-5 and 1e-3 for projector $f(\cdot)$ and $g(\cdot)$ in stage-1. During stage-2 and stage-3, we set the learning rate for both projectors and word embeddings as 2e-5, while 2e-4 for LoRA parameters ($r$ = 128 and $\alpha$ = 256). All experiments are conducted on 8 NVIDIA A100/H800 GPUs.

## A.2 INSTRUCTIONS FOR EACH TASK

The quality and diversity of instructions are essential in the training process. We manually write well-designed instructions for some tasks, combined with some templates in Time-IT (Ren et al., 2024). Table 9 lists the prompts for different tasks.

## A.3 EVALUATION PROCESS

For the evaluation of the temporal sentence grounding task, we directly input the prompt `["At which time interval in the video can we see < query > occurring?"]` in Table 9 to get the response `["From < start > to < end >"]`, and then calculate the predicted timestamps with the Equation (3) to get the IoU metrics.

For the evaluation of the dense video captioning task, we directly input the prompt `["Detect and report the start and end timestamps of activity events in the video, along with descriptions."]` in Table 9 to get the response `["From < start_1 > to < end_1 >, < caption_1 >. From < start_2 > to < end_2 >, < caption_2 >. ⋯ "]`, and then calculate the SODA_c (Fujita et al., 2020) and Meteor scores (Banerjee & Lavie, 2005).

For the evaluation of the visually-grounded VideoQA task, we adopt a two-round conversation evaluation as follows:

> **Round-1:**
> USER: Question: $< question >$. Options: $< options >$.
> ASSISTANT: Answer: $< answer >$.
> **Round-2:**
> USER: Provide the timestamps that correspond to your answer.
> ASSISTANT: From $< start >$ to $< end >$.

In the first round, we input the question and options into the model and get the answer. In the second round, we input the question, options, and predicted answer as the contexts, together with the prompt `["Provide the timestamps that correspond to your answer."]`, into the model to get the predicted timestamps. With both the predicted answers and timestamps, we can calculate the metrics including IoU, IoP, and Acc@GQA (Xiao et al., 2024).

For the evaluation of the Open-ended VideoQA and VCG-Bench, we employed GPT-3.5 turbo to juxtapose model outputs with ground truth data as introduced by Video-ChatGPT (Maaz et al., 2024b), subsequently computing both accuracy and a score. To ensure a fair and consistent compar-

ison with the results presented in Video-ChatGPT, we adopted the same prompt for our evaluations. For MVBench, we directly compute the accuracy of multiple-choice questions.

## A.4 VISUALIZATION PROCESS

We visualize the attention weights from the last layer of the LLM during the generation of a new temporal token. Since the full video representation consists of a total of $K \times (N_S + \frac{T}{K} \times N_T)$ tokens—where $T$, $K$, $N_S$, and $N_T$ denote the number of frames, number of segments, number of tokens per frame for the spatial stream, and number of tokens per frame for the temporal stream, respectively—we obtain an attention weight vector with the shape $[K \times (N_S + \frac{T}{K} \times N_T), 1]$. First, we discard the spatial stream portion, focusing only on the temporal information, which results in a new vector with the shape $[K \times \frac{T}{K} \times N_T, 1]$. We then reshape this vector to the form $[T, N_T, 1]$ and average it along the spatial dimension, yielding $[T, 1]$, which represents the final attention weights corresponding to each frame when generating a new token.

## A.5 DISTRIBUTION OF TEMPORAL TOKENS

We visualize the embeddings of the $M = 300$ temporal tokens to investigate their distribution in embedding space. We employ PCA (Abdi & Williams, 2010) to reduce the dimensionality of the temporal tokens to 1D, 2D, and 3D representations. For all reductions, we use the reduced values as coordinates, incorporating a gradient color scheme in which the color of the data points changes progressively with the token index, as illustrated in Figure 5. Our observations reveal that temporal tokens with similar indices tend to cluster together, exhibiting a continuous transition from tokens with smaller indices (light colors) to those with larger indices (darker colors).

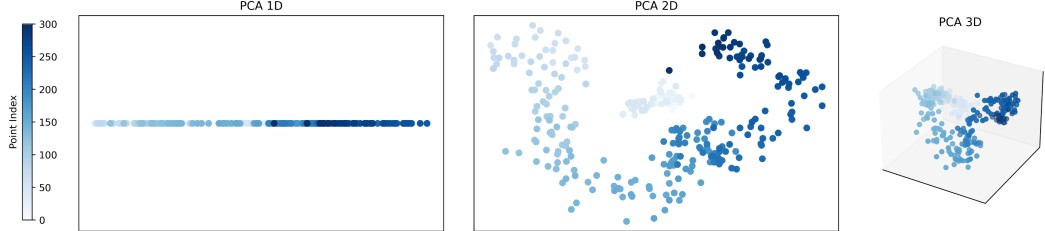

Figure 5: Visualization of temporal tokens with PCA.

Table 8: Prompts used to generate visually-grounded VideoQA samples with GPT-4.

**System**:
You are a good question generator. I need your help in generating question-answer pairs pertaining to the visual event descriptions. I have a video and I will provide you with descriptions of certain segments and their corresponding timestamps within this video. You need to consider these segments comprehensively based on the given description and timestamps, and select one segment which you think can provide a HIGH-QUALITY QUESTION. Based on the description of that segment, ask a question related to that segment, as well as one correct answer. Both the proposed answer and question should be consistent with the content of the give description. BE CAREFUL! Your proposed questions and answers should follow these rules:
(0) Avoid choosing the segment spanning across the whole video.
(1) The question you raised should include causal and temporal relationships as much as possible. Question types should be diverse including WHY, HOW, WHAT, WHERE, etc.
(2) NEVER involve anything that is not covered in the given descriptions.
(3) The answer should NEVER appear in your question.
(4) Your answer should be a phrase no more than 7 words. Keep your answers concise and accurate.

**Demonstrations**:

**User**:
video duration: 82.73 seconds
segment-1: [0.83, 19.86] A young woman is seen standing in a room and leads into her dancing.
segment-2: [17.37, 60.81] The girl dances around the room while the camera captures her movements.
segment-3: [56.26, 79.42] She continues dancing around the room and ends by laying on the floor.
**Response**:
chosen segment: segment-3
segment timestamps: [56.26, 79.42]
question: What did the girl do after she ended dancing?
answer: lay on the floor

. . .

(other in context demonstrations)
. . .

Table 9: Prompts used for different tasks.

**Prompts for Video-Caption Alignment**:
(1) "Describe the following video concisely.",
(2) "Provide a brief description of the given video clip.",
(3) "Offer a succinct explanation of the footage presented.",
(4) "Summarize the visual content of the following video.",
(5) "Give a short and clear explanation of the subsequent video clip.",
(6) "Share a concise interpretation of the video provided.",
(7) "Present a compact description of the clip's key features.",
(8) "Relay a brief, clear account of the video shown.",
(9) "Render a clear and concise summary of the video below.",
(10) "Write a terse but informative summary of the following video clip."

**Prompts for Temporal Sentence Grounding**:
(1) "When does $< query >$ happen in the video?",
(2) "At what time does the occurrence $< query >$ take place in the video?",
(3) "During which part of the video does $< query >$ occur?",
(4) "When in the video does the $< query >$ incident occur?",
(5) "At which moment does $< query >$ take place in the video?",
(6) "During which phase of the video does $< query >$ happen?",
(7) "When does the $< query >$ event occur in the video?",
(8) "At what time does $< query >$ occur in the video sequence?",
(9) "When does the $< query >$ situation take place in the video?",
(10) "At which time interval in the video can we see $< query >$ occurring?"

**Prompts for Dense Video Captioning**:
(1) "Localize a series of activity events in the video, output the start and end timestamp for each event, and describe each event with sentences.",
(2) "Detect and report the start and end timestamps of activity events in the video, along with descriptions.",
(3) "Pinpoint the time intervals of activity events in the video, and provide descriptions for each event.",
(4) "Can you compile a list of the activities and their timestamps featured in the video?",
(5) "I need you to scrutinize the video and catalog every event it contains, along with the timestamps."

**Prompts for Temporal Referring**:
(1) "What is happening from $< start >$ to $< end >$?",
(2) "What is taking place between $< start >$ and $< end >$?",
(3) "What events unfold between $< start >$ and $< end >$?",
(4) "What is happening during the period from $< start >$ to $< end >$?",
(5) "What occurs between $< start >$ and $< end >$?",
(6) "What is going on from $< start >$ to $< end >$?",
(7) "How do things progress from $< start >$ to $< end >$?",
(8) "Can you describe what happens from $< start >$ to $< end >$?",
(9) "Describe the events occurring between $< start >$ and $< end >$.",
(10) "Narrate the actions that unfold from $< start >$ to $< end >$."

