# OpenReview forum: "Grounded-VideoLLM: Sharpening Fine-grained Temporal Grounding in Video Large Language Models"
_ICLR.cc/2025/Conference — ICLR 2025 Conference Withdrawn Submission_

### Official Review · Reviewer_YFj1 · 2024-10-21

**Soundness:** 3
**Presentation:** 3
**Contribution:** 2
**Rating:** 3
**Confidence:** 5

**Summary:**

In this paper, the authors proposed Grounded-VideoLLM, which is a VideoLLM with superior capability in fine-grained video understanding which previous VideoLLMs lack.
To do so, a two-stream encoding network equipped with a conventional spatial encoder and a newly adopted temporal encoder is adopted.
Also, unified temporal tokens are used, which are tokens dedicated to timestamp representation, making it easier to represent timestep-related information.
With proposed methods and collected training dataset, including ones automatically created with the aid of GPT, Grounded-VideoLLM shows satisfactory performance on multiple fine-grained video understanding tasks, while also showing satisfactory performances on conventional coarse-grained tasks.

**Strengths:**

1. Proposed architectures (two-stream encoding, unified temporal tokens) and training scheme (three-stage training including GPT-generated dataset) seem natural
2. Performance of Grounded-VideoLLM is validated on multiple temporal video understanding tasks, showing promising results compared to previous methods. Notably, Grounded-VideoLLM also shows satisfactory performances on general video understanding tasks, but not only on fine-grained tasks.
3. The paper is well-written and easy to follow, allowing one to understand key points

**Weaknesses:**

1. For me, a temporal stream seems like a process of simply sampling more frames and pooling them in a way to preserve the temporal axis. Since the idea of pooling (or aggregating) information in a temporal axis for better temporal modeling is widely adopted, the novelty seems to be limited, although this work might be the first work to adopt the idea in VideoLLMs.
2. The idea of adopting temporal tokens dedicated to timestamp representation has already been proposed in previous works [1, 2]. Unified temporal tokens proposed in this work seem to have no significant difference from those works.
3. By adopting a temporal stream, a large number of tokens are additionally fed into LLMs, which may significantly increase the computation cost considering the decoder-only LLMs have time complexity quadratic to the number of tokens. A comparison of the number of tokens being fed into LLMs and the overall model throughput with SOTA VideoLLMs may be required.
4. Grounded-VideoLLM is built upon Phi3.5-Vision-Instruct, which is a relatively recent baseline. Since recent MLLMs may show better performance compared to MLLMs that baseline VideoLLMs have been built upon, performances of those VideoLLMs might not be comparable. Therefore, implementing the idea of Grounded-VideoLLM on MLLMs identical (or similar) to those of previous VideoLLMs might better show the effectiveness of the proposed methods.

**Questions:**

Below is the summary of the review.
Please consider the weaknesses and the summary below for rebuttal.

**Summary**

Overall, the proposed architectures and training scheme seem to be an effective way to enhance a VideoLLMs’ fine-grained understanding ability, validated on multiple benchmarks.

However, the technical contribution seems limited, since the idea of the temporal stream is already well-known in the field of video understanding, and [1, 2] has already proposed an idea identical to the unified temporal tokens.

Also, the authors should better discuss the additional inference cost induced by adopting a temporal stream.

**References**

1. Qian et al., Momentor: Advancing Video Large Language Model with Fine-Grained Temporal Reasoning, ICML 2024
2. Yang et al., Vid2Seq: Large-Scale Pretraining of a Visual Language Model for Dense Video Captioning, CVPR 2023

---

### Official Review · Reviewer_AyRD · 2024-11-02

**Soundness:** 3
**Presentation:** 3
**Contribution:** 2
**Rating:** 5
**Confidence:** 4

**Summary:**

This paper introduces Grounded-VideoLLM, an advanced Video Large Language Model designed to enhance fine-grained temporal grounding. The authors identify limitations in existing Video-LLMs related to temporal modeling and timestamp representation. They propose a solution by integrating a temporal stream for frame relationship encoding and discrete temporal tokens for timestamp representation. Their multi-stage training approach, along with the creation of a grounded VideoQA dataset, demonstrates that Grounded-VideoLLM significantly improves performance in fine-grained tasks, showcasing its potential as a versatile video assistant.

**Strengths:**

1. The paper is well written and easy to follow.
2. Grounded-VideoLLM performs well on both general Open-Ended VideoQA benchmarks and temporal grounding benchmark.

**Weaknesses:**

The main weekness of this paper is the lack of more fair comparison of both the design of two-stream encoder and temporal token with previous works to demonstrate its effectiveness.
1. **The effectiveness of two-stream encoder**: Does the two-stream design necessary? Temporal modeling is important for video time grounding tasks. However, ablations in Table 6 only consider to remove the temporal stream, what if removing the spatial stream? The experiments in Table 6 should consider two more settings:

(i) replacing the image encoder in baseline (1) and (2) with video encoders to extract features.

(ii) only use temporal stream, for key frame with 144 tokens per frame, while non-key frame with 8 tokens.

What is the advantages of the proposed two-stream design over the above two time modeling settings?

2. Table 6 only shows the performance on time grounding related benchmark, does the two-stream encoder also perform better on general Open-Ended VideoQA benchmarks?
3. **Comparison of different timestamp representations**. 1) Lack fair comparison between the proposed temporal tokens and **the plain texts** used in previous works. Actually, the choice of timestamp representation in video is similar to the bounding box representation in spatial grounding tasks, but in Table 2 of [1] has demonstrated that numerical representation seems to perform better than adding vocabulary. So I wonder does it have different conclusion on timestamp representation. 2) **Compare with other temporal modeling method**, e.g., Temporal Perception Module in Momentor [2].

[1] Shikra: Unleashing Multimodal LLM’s Referential Dialogue Magic

[2] Momentor: Advancing video large language model with fine-grained temporal reasoning.

**Questions:**

1.	It is better to provide the evaluation on other challenging video benchmarks like **long video datasets**, e.g., EgoSchema [3], MovieChat [4]

[3] Egoschema: A diagnostic benchmark for very long-form video language understanding.

[4] Moviechat: From dense token to sparse memory for long video understanding.

---

### Official Review · Reviewer_geR3 · 2024-11-03

**Soundness:** 3
**Presentation:** 3
**Contribution:** 2
**Rating:** 3
**Confidence:** 4

**Summary:**

The paper introduces Grounded-VideoLLM, a Video Large Language Model (Video-LLM) that is focused on fine-grained temporal grounding in videos. The proposed Grounded-VideoLLM approach utilizes a dual-stream architecture, where one stream encodes spatial features that represent the appearance and the other temporal features for motion. Furthermore, the authors propose to use temporal tokens to represent timestamps in the latent language space. Specifically, the model incorporates discrete temporal tokens, which allows for unified processing of temporal information and text within the model. The paper also leverages a progressive training scheme, which begins with simpler video-caption alignment and eventually advancing to complex temporal grounding tasks. Empirically, the authors demonstrate the effectiveness of their Grounded-VideoLLM approach, where it outperforms state-of-the-art approaches on multiple tasks including temporal sentence grounding, dense video captioning, and grounded video question answering.

**Strengths:**

1) In light of the gravitation towards large language models (LLMs) and their multimodal variants (MLLMs), the proposed approach addresses two really important research questions about the effectiveness of such models in understanding temporal information in the visual domain as well as generating more interpretable responses with regards to multimodal queries for videos.

2) In terms of clarity, the paper is well-written and motivated. The paper is well-organized, with clear explanations of the two-stream architecture, temporal tokens, and multi-stage training strategy. The model figures are informative and especially helpful in helping the reader to understand the different stages of the data curation process as well as the intuition behind each stage.

**Weaknesses:**

1) The proposed approach is relatively lacking in novelty, given its stated contributions. To begin, the benefits of the two-stream encoding mechanism used in the Grounded-VideoLLM approach have been well-documents by prior works such as the SlowFast and Two-stream Convolutional Networks. Additionally, the introduction of unified temporal tokens to convert continuous timestamp information into the latent space of word token representations in LLMs mirrors that of prior work [1, 2]. At  a high-level, the proposed approach appears to be simply combining different aspects of prior video understanding models. One possibility is to  provide a more detailed comparison with the above-mentioned work(SlowFast, Two-stream Convolutional Networks, LITA, and Vid2Seq) to help clarify the paper's novelty.

[1] LITA: Language Instructed Temporal-Localization Assistant, ECCV 2024
[2] Vid2Seq: Large-Scale Pretraining of a Visual Language Model for Dense Video Captioning, CVPR 2023

2) Given point (1) and the demonstrated performance gain of Grounded-VideoLLM over LITA and VTimeLLM in Table 4 on video question answering, it is not clear from the experiments if the gains are due to differences in the base language models used in the different approaches. It would be very helpful to include this ablation in the empirical analysis. For example, you can compare the proposed approach using the same base language model as LITA or VTimeLLM.

**Questions:**

See weaknesses.

---

### Official Review · Reviewer_pRgp · 2024-11-07

**Soundness:** 3
**Presentation:** 3
**Contribution:** 3
**Rating:** 6
**Confidence:** 5

**Summary:**

This paper introduce Grounded-VideoLLM, a novel Video-LLM adept at perceiving and reasoning over specific video moments
in a fine-grained manner. To solve the limitations of video LLM for fine-grained video understanding since they lack effective temporal modeling and timestamp representation, they propose  an additional temporal stream to encode the relationships between
frames and  discrete temporal tokens enriched with specific time knowledge to represent timestamps.To optimize the training of Grounded-VideoLLM, they employ a multi-stage training scheme, beginning with simple video-captioning tasks
and progressively introducing video temporal grounding tasks of increasing complexity.A grounded VideoQA dataset by an automatic annotation pipeline is build.

**Strengths:**

1. They introduce Grounded-VideoLLM, a Video-LLM adept at perceiving and reasoning over specific video moments.
2.They propose a step-by-step training strategy that progressively adapts an image-based MLLM into a robust Video-LLM, while curating a grounded VideoQA dataset to further enhance the temporal reasoning capability.

**Weaknesses:**

1.The table 2 shows the zero shot result. However, it seems bring ths knowledge about ActivityNet-Captions datasets in training stages. It should be explained detailed.

**Questions:**

1. Can you discuss the computational environment and time cost of finetuing this large models?

---

### Note · Authors · 2024-11-15

I have read and agree with the venue's withdrawal policy on behalf of myself and my co-authors.